# Maternal pandemic-related stress during pregnancy associates with infants' socio-cognitive development at 12 months: A longitudinal multi-centric study

**Sarah Nazzari**[1‡], **Serena Grumi**[2‡], **Giacomo Biasucci**[3], **Lidia Decembrino**[4], **Elisa Fazzi**[5,6], **Roberta Giacchero**[7], **Maria Luisa Magnani**[4], **Renata Nacinovich**[8,9], **Barbara Scelsa**[10], **Arsenio Spinillo**[11], **Elena Capelli**[2], **Elisa Roberti**[2], **Livio Provenzi**[1,2]*, **on behalf of the MOM-COPE Study Group**

1 Department of Brain and Behavioral Sciences, University of Pavia, Pavia, Italy, 2 Developmental Psychobiology, IRCCS Mondino Foundation, Pavia, Italy, 3 Pediatrics & Neonatology Unit, Ospedale Guglielmo da Saliceto, Piacenza, Italy, 4 Pediatric and Neonatal Unit, ASST Pavia, Vigevano, Italy, 5 Department of Clinical And Experimental Sciences, University of Brescia, Brescia, Italy, 6 Unit of Child and Adolescence Neuropsychiatry, Azienda Ospedaliera Spedali Civili of Brescia, Brescia, Italy, 7 Department of Pediatrics, Ospedale Civile di Lodi, Lodi, Italy, 8 Child and Adolescent Neuropsychiatry, Fondazione IRCCS San Gerardo dei Tintori Monza, Monza, Italy, 9 School of Medicine and Surgery and Milan Center for Neuroscience (NeuroMi), Università Bicocca, Milan, Italy, 10 Unit of Pediatric Neurology, Buzzi Children's Hospital, Milan, Italy, 11 Department of Obstetrics and Gynecology, IRCCS Policlinico San Matteo, Pavia, Italy

‡ SN and SG are considered as co-first authors to this work.
* livio.provenzi@unipv.it

## Abstract

### Background

Prenatal maternal stress is a key risk factor for infants' development. Previous research has highlighted consequences for infants' socio-emotional and cognitive outcomes, but less is known for what regards socio-cognitive development. In this study, we report on the effects of maternal prenatal stress related to the COVID-19 pandemic on 12-month-old infants' behavioral markers of socio-cognitive development.

### Methods

Ninety infants and their mothers provided complete longitudinal data from birth to 12 months. At birth, mothers reported on pandemic-related stress during pregnancy. At infants' 12-month-age, a remote mother-infant interaction was videotaped: after an initial 2-min face-to-face episode, the experimenter remotely played a series of four auditory stimuli (2 human and 2 non-human sounds). The auditory stimuli sequence was counterbalanced among participants and each sound was repeated three times every 10 seconds (Exposure, 30 seconds) while mothers were instructed not to interact with their infants and to display a neutral still-face expression. Infants' orienting, communication, and pointing toward the auditory source was coded micro-analytically and a socio-cognitive score (SCS) was obtained by means of a principal component analysis.

**Data Availability Statement:** The raw data related to the present publication are available from the Zenodo repository (url: 10.5281/zenodo.7516250).

**Funding:** This study is supported by funds from the Italian Ministry of Health (Cinque per Mille, 2017; Ricerca Corrente 2022) and Fondazione Roche Italia (Fondazione Roche per la Ricerca Indipendente 2020) to author LP. The funders had no role in study design, data collection, analysis, decision to publish, or preparation of the manuscript.

**Competing interests:** The authors have declared that no competing interests exist.

## Results

Infants equally oriented to human and non-human auditory stimuli. All infants oriented toward the sound during the Exposure episode, 80% exhibited any communication directed to the auditory source, and 48% showed at least one pointing toward the sound. Mothers who reported greater prenatal pandemic-related stress had infants with higher probability of showing no communication, $t = 2.14$ ($p = .035$), or pointing, $t = 1.93$ ($p = .057$). A significant and negative linear association was found between maternal prenatal pandemic-related stress and infants' SCS at 12 months, $R^2 = .07$ ($p = .010$), while adjusting for potential confounders.

## Conclusions

This study suggests that prenatal maternal stress during the COVID-19 pandemic might have increased the risk of an altered socio-cognitive development in infants as assessed through an observational paradigm at 12 months. Special preventive attention should be devoted to infants born during the pandemic.

## Introduction

Pregnancy is a period of complex development during which the fast-paced maturation of multiple neurophysiological and neurobehavioral domains is highly expectant on and susceptible to environmental stimuli [1, 2]. During this sensitive period, the stress experienced by pregnant women might be embedded into the progressing neurophysiological and neurobehavioral phenotype of the fetus [3, 4], contributing the long-lasting programming of infants' development possibly via neuroendocrine [5, 6], inflammatory [7], and epigenetic mechanisms [8, 9]. Infants of women reporting higher levels of stress during pregnancy have been found to show altered brain connectivity [10, 11] and to be at greater risk for a series of negative outcomes, including socio-emotional and stress dysregulation [12], cognitive deficit or delay [13], altered neurodevelopment [14], and even mental and physical health problems later in life [15, 16].

Meta-analytic evidence of studies conducted during the COVID-19 pandemic highlighted that pregnant women experienced high levels of distress during this global traumatic experience [17] and that this might associate with stress-related biological alterations for both mothers [18, 19] and infants [4, 20]. A US-based study reported that pregnant women who experienced high levels of pandemic-related stress had also adverse perinatal outcomes [21]. Notably, independent research studies are concordant in suggesting that pregnancy during the pandemic per se, but not exposure to SARS-CoV-2 infection, might significantly raise the risk for altered developmental outcomes [22, 23]. For example, higher risk of neurodevelopmental delay at age 6 months in infants born during the pandemic was observed in a large cohort study in New York City and the effect was not significantly associated with the direct exposure to COVID-19 infection [23]. Moreover, the study by Imboden et al. [24] reported a slight decrease in the communication domain scores among 12-month-old infants compared to a cohort born before the pandemic. These initial findings suggest that the negative impact of the pandemic is specifically linked to the prenatal exposure to the healthcare emergency and maternal stress. Noteworthy, Sperber et al. [25] did not report any significant effects of the pandemic on maternal mental health and infants' developmental outcomes among mother-infant dyads that experienced the onset of the pandemic during their first year of life. In this

perspective, pregnancy seems to represent a specific window of susceptibility for the effects of maternal pandemic-related stress on infants' development. Understanding pregnant women experiences of stress during the COVID-19 pandemic and assessing its effects during the first thousand days is critical to inform timely preventive interventions to protect infants' socio-emotional and socio-cognitive development [26].

Socio-cognitive behaviors include the capacity to orient, point, and communicate toward a third object and are meant to reach a mature stage toward the end of the twelfth month of life [27]. They are generally considered as half-way between socio-emotional skills–e.g., approach toward other humans, prosocial and collaborative behaviors–and cognitive abilities–e.g., recognizing others as separated, understanding others' goals, sharing a specific interest for an object in the environment [28]. For instance, by the end of the first year of life, infants show clear emergence of declarative pointing as a way to highlight their interest for something place in distal positions, to signal their state of mind to another interactive agent, thus organizing triadic exchanges and shared attention states that are key precursors of later theory of mind development [28–30]. Socio-cognitive behaviors can be easily observed and coded using video-taped of mother-infant interaction at this age, as they rely on a clear combination of gestures (e.g., pointing and reaching-out in the direction of the stimulus) and looking behaviors (e.g., orienting and fixating for appropriately prolonged time a specific target) [31]. These behaviors are critical indicators of potentially altered trajectories that might be, at least partly, programmed by prenatal exposure to adversities and that can lead to problems in psychological adjustment later in life [31]. Nevertheless, the majority of studies investigating the association between prenatal maternal stress and infant's development focused on indices of either cognitive [32, 33] or socio-emotional [34, 35] development, whereas the effect of prenatal maternal stress on infants' socio-cognitive development is largely unexplored.

In the present study, we investigated the association between maternal pandemic-related prenatal stress (PRS) and infants' socio-cognitive behaviors at 12 months. To this aim, we assessed specific socio-cognitive behaviors–i.e., orienting, pointing, and active communication–toward auditory human and non-human stimuli in 12-month-old infants born during the COVID-19 pandemic. Specifically, socio-cognitive behaviors were assessed in terms of their rate (how many infants displayed them at least once) and frequency (how often infants displayed them). Higher levels of PRS were hypothesized to associate with a lower rate and frequency of socio-cognitive behaviors in infants.

## Methods

### Participants

This study is part of the multi-centric and longitudinal Measuring the Outcomes of Maternal COVID-19-related Prenatal Exposure (MOM-COPE) research project. The complete protocol of the project is reported elsewhere [36]. The original sample of 320 mother-child dyads were enrolled between May 2020 and February 2021 from eight neonatal units in Northern Italy. Subjects were considered eligible to the study if mothers were at least 18 years old, in the absence of prenatal and perinatal diseases or injuries, if delivery was at term (i.e., from 37 + 0 to 41–6 weeks of gestation), and if mothers tested negative for SARS-CoV-2 at delivery. Here we report on a subset of mother-child dyads (N = 91) that provided complete data for perinatal and 12-month assessments. Dyads from this sample did not significantly differ from those who withdrawn in terms of stress experienced during pregnancy and socio-demographic characteristics, except for gestational age (t(318) = -2.687, p = .008; M = 39.90; SD 1.05 for the final sample; M = 39.54, SD 1.09 for the participants who dropped out) and maternal education (t(317) = -2.124, p = .034; M = 15; SD 3.35 for the final sample; M = 15.86, SD 2.91 for the

participants who dropped out). Complete sociodemographic characteristics of the participants who dropped out from the study as well as differences with the final study sample are reported in Supplementary Table 1 in S1 File.

### Ethical considerations

The study was approved by the Ethics Committees of Pavia (Italy) and the participating hospitals. All mothers provided written informed consent to participate in the study.

### Procedures and measures

Socio-demographic (maternal age, maternal education level) and neonatal (gestational age, birth weight, head circumference, neonatal length, Apgar score at minute 5) variables were obtained from medical records. Within 48 hours from delivery, mothers reported on their prenatal pandemic-related stress by filling in a self-report questionnaire (see S1 File). The questionnaire included six 5-point Likert scale items (1, not at all; 5, very much) on the emotional stress response to the COVID-19 emergency and an average maternal prenatal pandemic-related stress (PRS) score was obtained. The Cronbach's $\alpha$ of this scale was .834. Direct (own positivity with or without symptoms) or indirect (positivity, hospitalization or death of a relative or significant others) physical exposure to the SARS-CoV-2 virus was tracked using dichotomous items [36]. Only two women reported testing positive to the COVID-19 during pregnancy.

At infants' 12-month-age, mothers and infants participated from home to a remote videotaped face-to-face interaction. The remote observational procedure was structured according to recommendations from previous studies [37]. The online session was planned according to parents' availability and the videorecording started only when the infant was in a quiet alert state. Prior to the interaction, mothers were instructed to set the smartphone or the tablet in a position that allowed the experimenters to have a full side-view of both the interactive partners. For the entire duration of the interaction, both the mother and the experimenter switched-off their screen; the experimenter also muted her mic while sharing PC reproduced sound. The interactive task lasted 6 minutes (Fig 1). Mother and infant interacted face-to-face with no use of toys or pacifier for two minutes. Subsequently, one of four prerecorded auditory stimuli was played three times every 10 seconds (Exposure episode). Two auditory stimuli were human sounds–i.e., "Ciao" (Italian for "Hello") and "Che bello!" (Italian for "How nice!")–whereas the other two auditory stimuli were non-human–i.e., water and mixer sound. All infants were exposed to the entire set of four auditory stimuli, nonetheless the order of exposure was

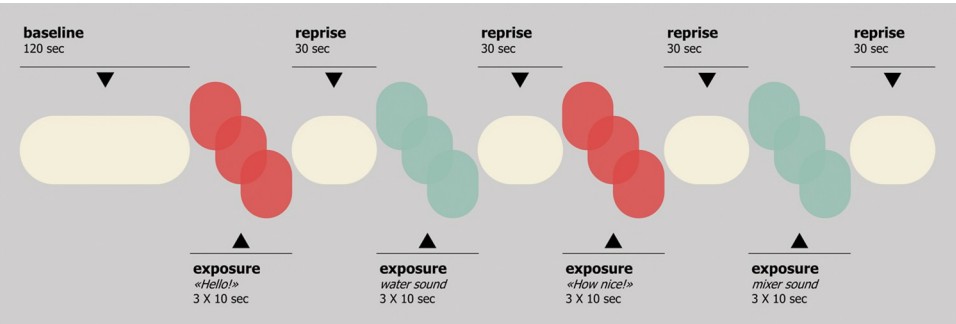

**Fig 1. Task overview. Note.** The order of the stimuli during the Exposure episodes was counterbalanced among subjects, keeping the alternation between human and non-human sounds. As the task was conducted with Italian mother-infant dyads, the human sounds were "Ciao" (for "Hello!") and "Che bello!" (for "How nice!").

counterbalanced among participants to avoid a sequence bias. During each Exposure episode, the mother maintained a still and neutral facial expression. After each 30-sec Exposure episode mother and infant resumed the face-to-face interaction (Reprise episode). Overall, a total of 102 mother-infant were recorded, but 11 videos were removed from the study because the task was not performed as required (in most of the cases the infant was crying in the highchair and was held by the mother for part of the task). All the videos had a frame and a video output quality that made them suitable for video-coding. For the purposes of the present study, specific infants' socio-cognitive behaviors–i.e., orienting, communication, and pointing–were micro-analytically coded during the Exposure episodes. Orienting was coded as any occurrence of infants' face and/or gaze clearly directed toward the source of the auditory stimuli. Communication was coded as any behavior of the infant clearly directed as communicating toward the source of the auditory stimuli, including waving bye with the hand, smiling, vocal production while looking at the sound source, etc. Pointing was coded as any clear gesture made with the index finger toward the source of the auditory stimuli. Two scores were obtained for each socio-cognitive behavior: a dichotomous index of display (1) or non-display (0) for the entire duration of the Exposure episodes and a continuous index (percentage of Exposure time). Two main trained coders, blind to all other study data, rated the videos independently. A subset of randomly selected videos (approximately 10%) were independently coded by a third independent coder and the percentage of inter-rater agreement was 99.48%.

## Plan of analysis

**Data reduction.** Two Principal Component Analysis (PCA) were used to reduce the complexity of behavioral data. All the socio-cognitive continuous indexes separately obtained for human (i.e., orienting, communication, pointing) and non-human (i.e., orienting, communication, pointing) auditory stimuli were included. A non-constrained Varimax solution with criterion eigenvalue greater than 1 revealed a three-component structure (see Table 1). The principal components (PCs) clearly clustered together each socio-cognitive behavior independently from the nature (human or non-human) of the auditory stimuli, thus yielding three principal components: PC-ORI (orienting, 27.6% variance), PC-COM (communication, 23.3% variance), and PC-POI (pointing, 27.5% variance). A further constrained one-factor PCA solution with Varimax rotation was carried to test the presence of an over-arching socio-cognitive score (SCS). The constrained solution revealed one PC explaining 41.3% of total variance (Table 2).

**Main analyses.** Separate independent-sample $t$-tests were used to assess the presence of statistically significant differences in maternal prenatal PRS between infants who showed or not each of the socio-cognitive behaviors (i.e., orienting, communication, pointing). Bivariate

**Table 1. Non-constrained Varimax solution for socio-cognitive behaviors principal component analysis (N = 91).**

|  | Component | | | |
|---|---|---|---|---|
|  | 1, PC-ORI | 2, PC-POI | 3, PC-COM | Uniqueness |
| Orienting to non-human sound | 0.908 |  |  | 0.162 |
| Orienting to human sound | 0.836 |  |  | 0.270 |
| Pointing to human sound |  | 0.886 |  | 0.204 |
| Pointing to non-human sound |  | 0.870 |  | 0.185 |
| Communication to human sound |  |  | 0.920 | 0.151 |
| Communication to non-human sound |  |  | 0.703 | 0.322 |

Note. Loadings below .40 are hidden.

**Table 2. Constrained Varimax solution for socio-cognitive behaviors principal component analysis (N = 91).**

| | Component | |
| --- | --- | --- |
| | 1, SCS | Uniqueness |
| Pointing to non-human sound | 0.734 | 0.462 |
| Communication to non-human sound | 0.720 | 0.481 |
| Pointing to human sound | 0.639 | 0.592 |
| Orienting to human sound | 0.634 | 0.598 |
| Orienting to non-human sound | 0.626 | 0.608 |
| Communication to human sound | 0.469 | 0.780 |

Note. Loadings below .40 are hidden.

Pearson's correlations were used to assess the presence of statistically significant linear associations between maternal prenatal PRS and infants' socio-cognitive development (i.e., PC-ORI, PC-COM, PC-POI, SCS). The Benjamini-Hochberg algorithm was used to adjust for multiple comparison bias. Lastly, separate hierarchical regression models were used to further test the effect of PRS on infants' socio-cognitive development continuous measures while adjusting for potential infant and maternal confounders. All the analyses were performed using Jamovi 2.2.5.0 for Windows.

## Results

The study sample characteristics are reported in Table 3. For full description of the original sample of 320 dyads see Supplementary Table 1 in S1 File. The sample was balanced for infants' sex (44 females, 48.4%, 47 males, 51.6%). All infants oriented toward the sound during the Exposure episode, 80% exhibited any communication directed to the auditory source, and 48% showed at least one pointing toward the sound. The distribution of infants' continuous socio-cognitive measures (i.e., PC-ORI, PC-COM, PC-POI, and SCS) is reported in Fig 2.

Infants who did not display communication toward to the auditory source had mothers who reported greater prenatal PRS, $t(89) = 2.14$, $p = .035$, compared to communicating counterparts (Fig 3). Infants who did not display pointing toward to the auditory source had mothers who tended to report greater prenatal PRS, $t(89) = 1.93$, $p = .057$, compared to pointing counterparts (Fig 3). Significant linear associations emerged for maternal prenatal PRS and infants' PC-COM, $r = -.26$, $p = .015$ (Fig 4), as well as with infants' SCS, $r = -.29$, $p = .006$ (Fig 5). No significant associations emerged for PRS with both infants' PC-ORI and PC-POI.

The hierarchical regression models included: infant sex, gestational age, and birth weight at step 1; maternal age and maternal educational level at step 2; PRS at step 3. Final models included all variables tested at Step 1, 2 and 3. When testing the model on infants' SCS (see

**Table 3. Sample description (N = 91).**

| | Mean | SD | Minimum | Maximum |
| --- | --- | --- | --- | --- |
| Gestational age (weeks) | 39.901 | 1.055 | 37.00 | 42.0 |
| Birth weight (grams) | 3382.593 | 421.426 | 2480.00 | 4435.0 |
| Head circumference (cm) | 34.247 | 1.313 | 30.00 | 39.0 |
| Neonatal length (cm) | 50.593 | 2.015 | 46.00 | 56.0 |
| Apgar at minute 5 | 9.879 | 0.360 | 8.00 | 10.0 |
| Maternal age (years) | 33.154 | 4.681 | 18.00 | 46.0 |
| Maternal education (years of study) | 15.857 | 2.912 | 8.00 | 22.0 |

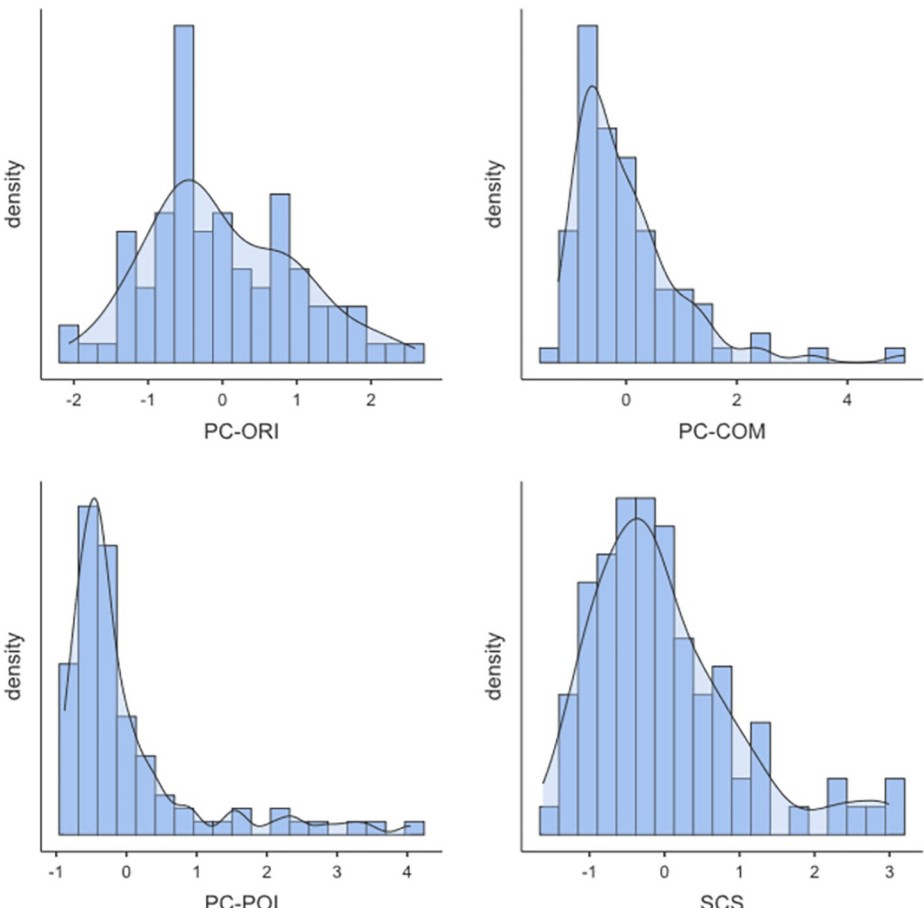

**Fig 2. Distribution of infants' socio-cognitive development continuous measures: Orienting to auditory source, PC-ORI; pointing to auditory source, PC-POI; communication to auditory source, PC-COM; cumulative socio-cognitive score, SCS.**

Table 4A), a significant $\Delta R^2$ = .07 ($p$ = .010) was obtained only for step 3 which highlighted a significant effect of PRS on infants' socio-cognitive development at 12 months, while controlling for potential confounders. When looking at specific socio-cognitive scores, the model was significant for infants' PC-COM only at step 3, yielding a $\Delta R^2$ = .06 ($p$ = .017) and a significant effect of PRS on infants' PC-COM at 12 months (see Table 4D). The model was not significant when testing infants' PC-ORI (see Table 4A) and PC-POI (see Table 4C).

## Discussion

The present study is among the first to suggest the presence of a long-lasting association between maternal stress related to the experience of the pandemic during pregnancy and infants' socio-cognitive development at 12 months of age. While all the infants enrolled in this study showed appropriate orienting to auditory stimuli source during the interactive task, the majority of them (80%) also showed active communications directed to the sound while only half of the sample (48%) displayed index finger pointing gesture. These findings confirm that deictic gesture is still developing at the end of the first year of life and they extend previous evidence from visual stimuli [28, 38] to auditory ones [39].

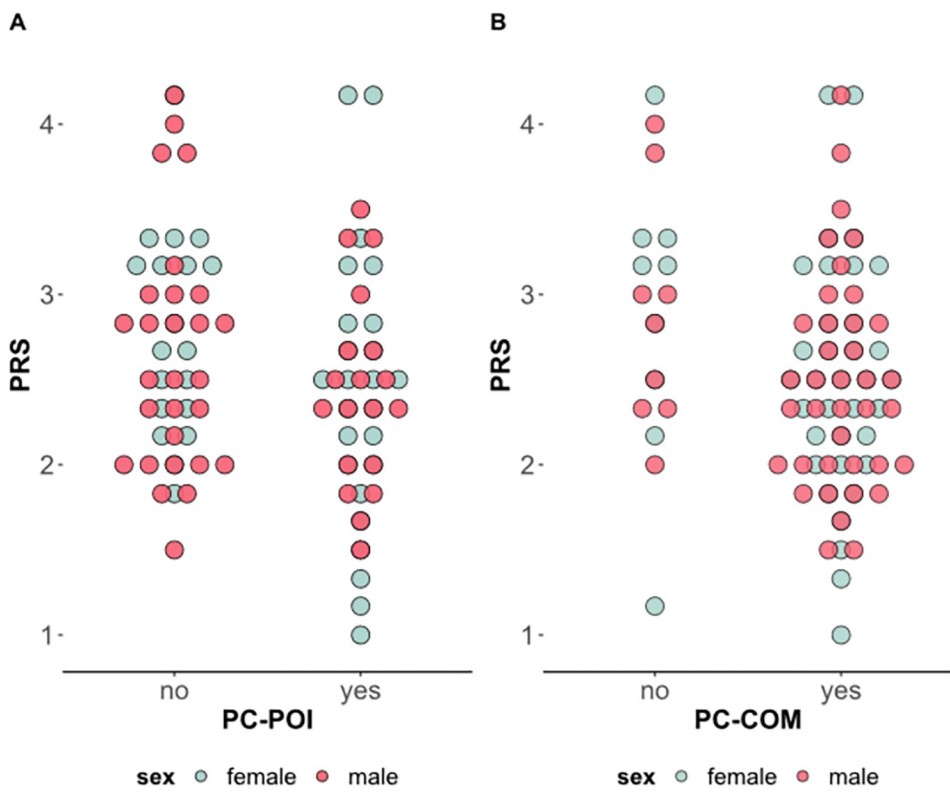

**Fig 3. Maternal pandemic-related stress (PRS) during pregnancy by infants' display of communication and pointing toward auditory stimuli at 12-month-age.**

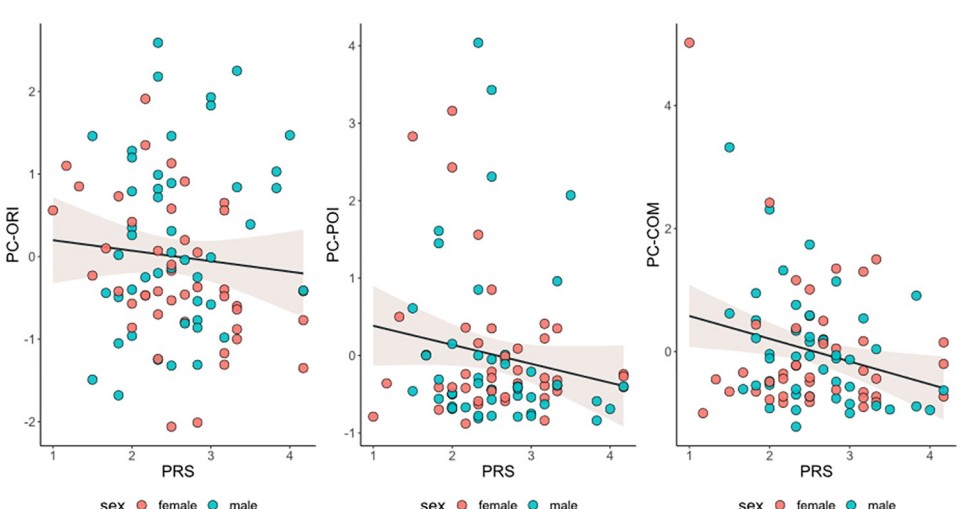

**Fig 4. Linear association between maternal pandemic-related stress (PRS) during pregnancy and infants' orienting (PC-ORI), pointing (PC-POI), communication (PC-COM) at 12-month-age. Note.** All outcome variables on the y-axis are standardized (mean = 0, sd = 1). In regression model predicting PC-ORI, standardized estimate for PRS = -0.09, p = 0.41; In regression model predicting PC-POI, standardized estimate for PRS = -0.16, p = 0.16; In regression model predicting PC-COM, standardized estimate for PRS = -0.26, p = 0.02.

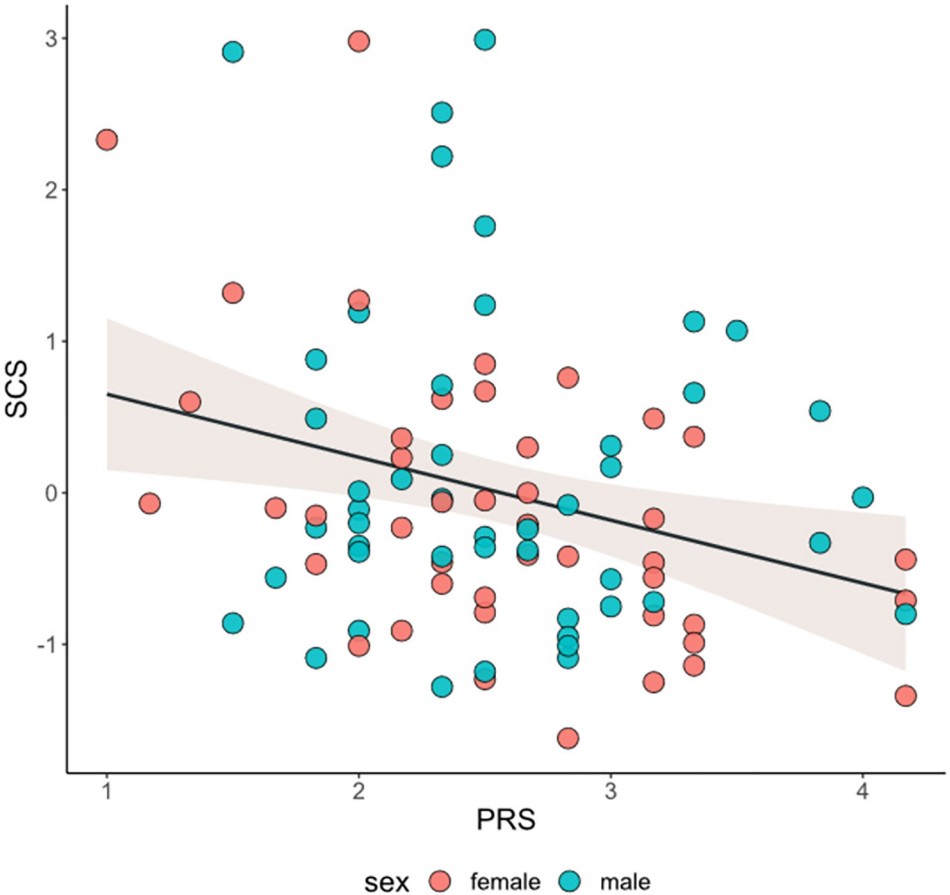

**Fig 5. Linear association between maternal pandemic-related stress (PRS) during pregnancy and infants' over-arching socio-cognitive score (SCS) at 12-month-age. Note.** Infant SCS scores are standardized (mean = 0, sd = 1). Standardized estimate for PRS = -0.28, p = 0.01.

Notably, variability in socio-cognitive behaviors appeared to be at least partially associated with maternal stress exposure during pregnancy: those infants who did not show pointing and active communications were also exposed to higher levels of prenatal pandemic-related stress. Evidence on the long-lasting effects of prenatal exposure to the COVID-19 pandemic on infant development across the first postnatal year is still scarce. We previously reported on the association between PRS and infant temperament at 3 months of age [22, 40]. Furthermore, prenatal maternal distress during the pandemic was associated with infant brain connectivity at 3 months of age [41] and with poorer socio-emotional development at 2 months of age [42]. Consistently, infants born during the pandemic had higher risk of neurodevelopmental delays at 6 months [23] and lower communication scores at 12 months [24], as compared with historical cohorts born before the pandemic. Current findings suggest that the potential negative effects of PRS per se–even in absence of antenatal exposure to SARS-CoV-2 infection–on offspring's development are likely to persist until 12 months after childbirth. Furthermore, while previous research on the effects of maternal prenatal stress mainly reported on emotional [43, 44] and cognitive [13, 45] infant outcomes, here we suggest that a precocious marker of altered developmental trajectory as consequence of adversity exposure during pregnancy might be traced in the early delay or deficit in specific socio-cognitive behaviors. Mechanisms

**Table 4. Hierarchical regression models testing the effect of maternal PRS on infants' socio-cognitive development at 12 months (N = 91).**

| **A. SCS** | | | | | | | |
|---|---|---|---|---|---|---|---|
| Step | $R^2$ | $\Delta R^2$ | Cross-step sig. | Predictors | Standardized coefficient | 95% Confidence Interval | Predictor sig. |
| 1 | 0.02 | | | Sex | 0.24 | [-0.17,0.66] | 0.25 |
| | | | | Gestational age | -0.02 | [-0.26,0.23] | 0.89 |
| | | | | Birth weight | 0.06 | [-0.17,0.30] | 0.59 |
| 2 | 0.03 | 0.01 | 0.57 | Maternal age | -0.02 | [-0.24,0.20] | 0.88 |
| | | | | Maternal education | 0.08 | [-0.13,0,30] | 0.44 |
| 3 | 0.10 | 0.07 | 0.01 | PRS | -0.28 | [-0.49,-0.07] | 0.01 |
| **B. PC-ORI** | | | | | | | |
| Step | $R^2$ | $\Delta R^2$ | Cross-step sig. | Predictors | Standardized coefficient | 95% Confidence Interval | Predictor sig. |
| 1 | 0.05 | | | Sex | 0.41 | [-0.02,0.84] | 0.06 |
| | | | | Gestational age | 0.03 | [-0.22,0.28] | 0.78 |
| | | | | Birth weight | -0.04 | [-0.28,0.20] | 0.75 |
| 2 | 0.06 | 0.01 | 0.66 | Maternal age | -0.11 | [-0.12,0.33] | 0.34 |
| | | | | Maternal education | -0.01 | [-0.24,0,21] | 0.9 |
| 3 | 0.07 | 0.01 | 0.41 | PRS | -0.09 | [-0.31,0.13] | 0.41 |
| **C. PC-POI** | | | | | | | |
| Step | $R^2$ | $\Delta R^2$ | Cross-step sig. | Predictors | Standardized coefficient | 95% Confidence Interval | Predictor sig. |
| 1 | < .01 | | | Sex | -0.02 | [-0.45,0.42] | 0.93 |
| | | | | Gestational age | -0.01 | [-0.26,0.24] | 0.92 |
| | | | | Birth weight | -0.03 | [-0.28,0.22] | 0.78 |
| 2 | 0.01 | 0.01 | 0.81 | Maternal age | -0.04 | [-0.27,0.19] | 0.72 |
| | | | | Maternal education | 0.04 | [-0.19,0,27] | 0.73 |
| 3 | 0.03 | 0.02 | 0.16 | PRS | -0.16 | [-0.38,0.06] | 0.16 |
| **D. PC-COM** | | | | | | | |
| Step | $R^2$ | $\Delta R^2$ | Cross-step sig. | Predictors | Standardized coefficient | 95% Confidence Interval | Predictor sig. |
| 1 | 0.04 | | | Sex | 0.03 | [-0.39,0.44] | 0.90 |
| | | | | Gestational age | -0.06 | [-0.29,0.18] | 0.65 |
| | | | | Birth weight | 0.22 | [-0.02,0.45] | 0.07 |
| 2 | 0.07 | 0.03 | 0.22 | Maternal age | -0.11 | [-0.33,0.11] | 0.32 |
| | | | | Maternal education | 0.14 | [-0.08,0,35] | 0.21 |
| 3 | 0.13 | 0.06 | 0.02 | PRS | -0.26 | [-0.46,-0.05] | 0.02 |

Note. SCS, socio-cognitive score; PC-ORI, orienting; PC-POI, pointing; PC-COM, communication; PRS, pandemic-related stress.

underlying the observed associations are still unknown, but likely involved neuroendocrine, epigenetic, and environmental pathways. For example, pandemic-related stressors have been associated with greater maternal hair cortisol levels [19], thus suggesting a significant activation of maternal biological stress response systems during the pandemic, possibly leading to greater fetal exposure to maternal stress hormones and affecting later development. Furthermore, infants antenatally exposed to higher levels of PRS were found to have higher levels of SLC6A4 gene methylation at birth, which, in turn, was associated with later infant's temperament [40]. Additionally, higher levels of DNA methylation of stress-related genes were previously reported in infants prenatally exposed to the COVID-19 pandemic [46], thus suggesting that epigenetic mechanisms might play a role in mediating the embedding of the prenatal exposure to the pandemic on infants' development. Lastly, maternal postnatal distress was found to partially mediate the effects of maternal PRS respectively on infants' socio-emotional development at 2 months [42] and regulatory capacity at 3 months [22]. Likewise, the quality

of mother-infant postnatal relationship might play a mediating or moderating [20, 47] role in the observed associations. Noteworthy, Hane and colleagues [48] recently showed that infants of emotionally connected dyads showed more approach-seeking behaviors and social engagement as well as a specific autonomic activation patterns as compared to those from non-connected dyads during the face-to-face still-face task. Future studies would benefit from the inclusion of physiological measures along with behavioral markers in order to elucidate whether the observed differences in children's socio-cognitive outcomes possibly reflect differences in autonomic responding, in line with the novel hypothesis of an autonomic socio-emotional "reflex" pathway [49]. This would contribute shedding light on the mechanisms underlying the effects of PRS on infant development.

Interestingly, the present study further supports the hypothesis of a linear association between prenatal stress and infants' early socio-cognitive development at 12 months. This association remained significant even after controlling for potential infants and maternal confounders. The effect was specifically evident for infants' tendency to produce active communications (e.g., waving by, smiling, vocalizing) toward an auditory stimulus while interacting with their mother. A significant relationship between prenatal stress exposure and child outcomes has often been reported [50, 51]. The current findings extend this evidence by showing that even the association between PRS and infants' socio-cognitive development behaves in a linear fashion, with important clinical implications. Assessment of maternal psychological well-being during the pandemic is fundamental as PRS is likely to have a small though detectable impact even at subclinical levels. Furthermore, formal and informal support should be offered to expectant families, even more during a pandemic, in order to protect maternal and infant perinatal mental health.

The study has limitations. First, the sample size is relatively low which also depends on the longitudinal and observational nature of the project which, especially during the pandemic time, has resulted in remarkable sample attrition (only 28.44% of the dyads of the original sample took part in the observational task planned at 12 months). Additionally, potential systematic patterns of missingness limit generalizability of the current findings. In particular, while small differences in gestational age and maternal education were found among dyads who attended the follow-up and dyads who withdrew, only variables collected at birth were available for comparison among the two groups, thus possible selective attrition on unobserved variables remained an issue. Second, we adapted previously available tasks to assess infants' socio-cognitive behavior remotely by developing a new observational procedure. The accumulating experiences during the pandemic period about remote data collection [37] informed the present procedure and allowed us to obtain a high number of videos suitable for remote coding. However, the findings derived from the remote collection in a home setting are not directly comparable to those from lab settings and should be considered descriptive and not clinically informative. Third, the prenatal stress questionnaire was developed ad hoc for this study prioritizing sensitivity to the specific and unprecedented nature of COVID-19 emergency over measure standardization. Lastly, the correlational design prevents us from establishing causal links.

## Conclusions

The present study suggests that infants who were born during the pandemic might be at risk for an altered trajectory in socio-cognitive development and that this risk might be at least partially associated with the extent of maternal pandemic-related prenatal stress. These findings add to previous reports on socio-emotional and cognitive developmental risk in infants' born during the pandemic. As such, we suggest that this population should be specifically followed-up to inform smarter preventive care during and after the pandemic period.

## Supporting information

**S1 File.**
(DOCX)

## Acknowledgments

The authors are thankful to the colleagues of the MOM-COPE Study Group: G. Bensi, R. Bonini, R. Borgatti, A. Cavallini, R. Falcone, B. Gardella, G. Kullmann, V. Manfredini, F. Masoni, S. Orcesi, D. Pantaleo, G. Pettenati, B. Pietra, C. Pisoni, F. Prefumo, V. Spartà.

## Author Contributions

**Conceptualization:** Livio Provenzi.

**Data curation:** Sarah Nazzari, Elena Capelli, Elisa Roberti, Livio Provenzi.

**Formal analysis:** Elena Capelli, Elisa Roberti, Livio Provenzi.

**Funding acquisition:** Livio Provenzi.

**Investigation:** Serena Grumi, Giacomo Biasucci, Lidia Decembrino, Elisa Fazzi, Roberta Giacchero, Maria Luisa Magnani, Renata Nacinovich, Barbara Scelsa, Arsenio Spinillo, Elisa Roberti, Livio Provenzi.

**Methodology:** Serena Grumi, Livio Provenzi.

**Resources:** Elena Capelli.

**Supervision:** Sarah Nazzari, Livio Provenzi.

**Visualization:** Sarah Nazzari, Elena Capelli.

**Writing – original draft:** Sarah Nazzari, Serena Grumi.

**Writing – review & editing:** Giacomo Biasucci, Lidia Decembrino, Elisa Fazzi, Roberta Giacchero, Maria Luisa Magnani, Renata Nacinovich, Barbara Scelsa, Arsenio Spinillo, Elena Capelli, Elisa Roberti, Livio Provenzi.

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
