## [Decision Letter · Decision Letter 0]

14 Nov 2022

PONE-D-22-23782Maternal pandemic-related stress during pregnancy associates with infants’ socio-cognitive development at 12 months: A longitudinal multi-centric studyPLOS ONE

Dear Dr. Provenzi,

Thank you for submitting your manuscript to PLOS ONE. After careful consideration, we feel that it has merit but does not fully meet PLOS ONE’s publication criteria as it currently stands. Therefore, we invite you to submit a revised version of the manuscript that addresses the points raised during the review process.

 Please provide in the revised manuscript where the data may be found (doi). In the original submission the authors only stated: "All the raw data related to the present publication are available from the XXX database (doi: XXX)."

We look forward to receiving your revised manuscript.

Kind regards,

Claudine Irles, Ph.D.

Academic Editor

PLOS ONE

Journal Requirements:

3. One of the noted authors is a group or consortium “MOM-COPE Study Group”. In addition to naming the author group, please list the individual authors and affiliations within this group in the acknowledgments section of your manuscript. Please also indicate clearly a lead author for this group along with a contact email address.

4. Please upload a copy of Supplementary File S1 which you refer to in your text on page 5 and 6.

Reviewers' comments:

Reviewer's Responses to Questions

**Comments to the Author**

1. Is the manuscript technically sound, and do the data support the conclusions?

Reviewer #1: Yes

Reviewer #2: Partly

2. Has the statistical analysis been performed appropriately and rigorously? 

Reviewer #1: Yes

Reviewer #2: No

3. Have the authors made all data underlying the findings in their manuscript fully available?

Reviewer #1: Yes

Reviewer #2: Yes

4. Is the manuscript presented in an intelligible fashion and written in standard English?

Reviewer #1: Yes

Reviewer #2: Yes

5. Review Comments to the Author

Reviewer #1: the manuscript technically sound, and do the data support the conclusions. I am not sure but I am assuming the data has been made available. the manuscript is well written in a way that makes the subject very understandable.

Reviewer #2: The present study evaluates the associations between prenatal maternal stress during the pandemic and infant socio-cognitive functioning at 12-months postpartum. The authors find that infants of mothers who reported higher pandemic-related stress during pregnancy were less likely to point and demonstrate communication when presented with auditory stimuli. The study is clearly written, and has the potential to make a valuable contribution to the literature.

However, several points should be addressed. Overall, I take issue with the analytic approach (t-tests and correlations) and the cherry picking of results (frequency of socio-cognitive behaviors are not reported or interpreted, only the presence of them). Critically, the content of the pandemic stress measure is not presently available (the manuscript says it is available in the Supplement, but I do not have access to it). My recommendation for publication relies heavily on the availability of these items, since it is not a validated scale.

Intro

• The authors cite evidence related to increases in depression during the pandemic, but the main predictor in this paper is perceived stress. Clinical symptoms of mental health disorders are different from perceived stress levels, and we should avoid conflating the two constructs.

Method

• Need more information about how the domains of socio-cognitive functioning were coded. Were there multiple coders, and how was reliability assessed?

• The authors mention there was “remarkable sample attrition,” thus they only report results on infants with complete data. What was the original sample size, and what are the patterns of missing data? It is likely that large amounts of attrition will bias results. The authors should also consider using some method of imputation to reduce bias.

• Limitations of the remote interaction task should be considered. Are there any other published studies that have coded a remote parent-child interaction task? This is a novel method, and any evidence linking remote recordings to in-person recordings should be cited. Additionally, I would appreciate more details on the remote interaction (overall quality of the recordings, percent of recordings that were unusable, instructions given to the parent, etc) as this may be of use to future researchers.

Results

• An eligibility criteria for the study required mothers to test negative for COVID at delivery. The authors cite important evidence by Shuffrey et al, which reports neurodevelopmental delays of infants born to mothers who reported a COVID infection during pregnancy. Therefore, the authors should conduct a robustness check on mothers who tested positive for COVID during pregnancy with infant SCS scores (frequency & rate) if possible as a supplemental analysis

• The authors collected data on not just the occurrence, but also the frequency by which infants displayed these pointing/communication behaviors during the auditory task. These results are missing from the manuscript

• The authors test for correlations between potential confounds but do not report the results of these tests. Why not just control for confounds using linear regression? Even if the covariates are not significantly related to either the predictor or outcome variable, they are theoretically important to the outcome and accounting for them will result in a more precise estimate. I recommend the authors use logistic regression for the dichotomous indicators to obtain odds ratios (and linear regression with standardized predictor variables for the continuous outcomes), and include covariates for the demographic characteristics listed

• Did you account for age of the infant at time of the assessment? This should also be included as a covariate, as there might be differences in developmental timing of pointing/communication abilities by age

• What is the socioeconomic spread of the sample? Maternal education is reported, but is there any further information such as income?

• In observing Figure 2, there appears to be an outlier (high PC-COM score). Are these results robust to outliers? This should be reported in the supplement, given the small sample

Discussion:

• The discussion begins with the sentence “the present study is among the first to suggest the presence of a long-lasting effect of maternal stress…”. This terminology is again used in the next paragraph “…socio-cognitive behaviors appeared to be at least partially affected by maternal stress exposure during pregnancy.” This implies causation.

• This study would be helpful to cite in either the intro or discussion – it is not correlational, which provides more compelling evidence: Imboden, A., Sobczak, B. K., & Griffin, V. (2021). The impact of the COVID-19 pandemic on infant and toddler development. Journal of the American Association of Nurse Practitioners. https://doi.org/10.1097/JXX.0000000000000653

• This recent study also is a natural experiment: Sperber, J. F., Hart, E. R., Troller-Renfree, S. V., Watts, T. W., & Noble, K. G. (2022). The effect of the COVID-19 pandemic on infant development and maternal mental health in the first 2 years of life. Infancy. https://doi.org/10.1111/infa.12511

• I challenge the authors to remove terminology such as “suboptimal” development, as it assigns value judgements to developmental trajectories. It is possible (and there is a large body of literature arguing this) that developmental differences may be “suboptimal” according to traditional Western contexts, but may be adaptive and advantageous in adverse contexts. I encourage the writers to remove value judgements of developmental trajectories, and explore this idea further – how might socio-cognitive difference due to early stress exposure be adaptive in certain contexts?

6. PLOS authors have the option to publish the peer review history of their article (what does this mean?). If published, this will include your full peer review and any attached files.

Reviewer #1: No

Reviewer #2: No

---

## [Author Response · Author response to Decision Letter 0]

9 Jan 2023

REVIEWER #1 

1) The manuscript technically sound, and do the data support the conclusions. I am not sure but I am assuming the data has been made available. The manuscript is well written in a way that makes the subject very understandable. In this study, the authors have assessed possible effects of COVID related prenatal maternal stress on infant socio-cognitive development. They found a significant negative linear association between maternal stress and the Infant’s socio-cognitive score at 12 months and conclude that findings suggest preventive measures should be taken during the pandemic. The authors are to be congratulated on a very interesting paper. The study is well-designed, the manuscript is well-written and the findings add important evidence about the impact of COVID on human health. This paper fits the scope, and meets the criteria for publication in Plos One and in my opinion should be accepted.

REPLY. We wish to thank Reviewer 1 for appreciating this work and for highlighting its implications. The raw data will be made available on permanent third-party archive (Zenodo). We added in text a data availability statement to clarify this aspect.

2) I do have some suggestions that I think could make the paper even more interesting. Introduction: The still-face is a well-validated paradigm designed to measure how an infant responds to a maternal stressor. The author’s have introduced a novel auditory stimulus during the still-face segment, coded orienting behaviors of the infant, and interpreted the results according to accepted orienting theory, which focuses on the cortically controlled orienting “response” phenomenon. The author’s might want to cite a recent paper that extensively reviews the orienting reflex theory, from Pavlov to present (Ludwig and Welch 2022 Frontiers in Psych) and concludes by identifying a novel autonomic socioemotional “reflex” (ASR) pathway. To that point, a recent still-face study coded orienting as a socioemotional reflex behavior pertaining to the mother-infant “relationship” – validating a new mother-infant emotional connection construct (see Hane AA et al. 2016 Act Paediatr).

REPLY. We thank Reviewer 1 for these suggestions. The suggested references have now been included in the manuscript to broaden the discussion about possible mechanisms underlying the observed associations in this study.

3) Discussion: The authors might want to refer to other studies that showed that maternal stress before, during and after pregnancy, whatever the cause, has profound effects on the development and lifelong function of the infants neurocognitive development. (Sharma etal 2022 Clin Epigenetics). The authors might want to cite some studies that looked at perinatal stress that emerged after 9-11. (Levendosky et al 2021 BMJ Open) (Van Sjelghem etal 2022 Early Hum Dev), which are similar to those emerging from Pandemic stress.

REPLY. Thank you for these suggestions, we have added in the introduction and discussion the reference to the study by Sharma et al. (2022) to strengthen the rationale about how prenatal adversities may be embedded into the progressing neurophysiological and neurobehavioral phenotype of the fetus.

4) The authors note in Discussion; “The effect was specifically evident for infants’ tendency to produce active communications (e.g., waving by, smiling, vocalizing) toward an auditory stimulus while interacting with their mother.” This observation deserves a paragraph to consider possible explanations of the orienting reflex data in this study. For instance, ASR theory predicts that those infants emotionally connected to the mother will display approach behaviors (i.e., attend to, orient to, etc.), while those not emotionally connected would display lessor or avoidant orienting behaviors. While mother-infant connection was not assessed in this study, it would be interesting in future studies to test the ASR hypothesis by conducting post-hoc analyses of mother-infant video interactions collected in this study using the Welch Emotional Connection Screen (WECS) to assess infant orienting to the mother and their emotional connection. Going forward, should the authors do further longitudinal follow-up studies, autonomic measures, such as HR and vagal tone could provide clinically relevant information from this study.

REPLY. We thank Reviewer 1 for making this point. Following also comment 2, we have now broadened the discussion of our findings to include possible environmental mechanisms encompassing the quality of mother-infant relationship that might underline the observed associations.

REVIEWER #2:

The present study evaluates the associations between prenatal maternal stress during the pandemic and infant socio-cognitive functioning at 12-months postpartum. The authors find that infants of mothers who reported higher pandemic-related stress during pregnancy were less likely to point and demonstrate communication when presented with auditory stimuli. The study is clearly written, and has the potential to make a valuable contribution to the literature.

REPLY. We wish to thank Reviewer 2 for carefully reading our manuscript and for his/her useful comments.

However, several points should be addressed. Overall, I take issue with the analytic approach (t-tests and correlations) and the cherry picking of results (frequency of socio-cognitive behaviors are not reported or interpreted, only the presence of them). Critically, the content of the pandemic stress measure is not presently available (the manuscript says it is available in the Supplement, but I do not have access to it). My recommendation for publication relies heavily on the availability of these items, since it is not a validated scale.

REPLY. We thank the Reviewer for raising these points. We addressed the Reviewer’s comments to the analytical approach in response to comments number 6 and 7.

Regarding to the ad-hoc pandemic-related stress measures, we regret that the reviewer did not have access to the supplementary material that we submitted together with the manuscript. We report also here the list of the ad hoc-items developed to assess physical exposure to COVID-19 (A) and pandemic related stress (B) during pregnancy. As noted by the Reviewer 2, the pandemic-related stress scale was specifically designed to assessed maternal stress during the COVID-19 pandemic and thus was not previously validated. Nevertheless, as now reported in the manuscript, the scale has a good internal consistency (Cronbach’s α =.834) and was already used in many previous published studies, such as:

• Grumi, Provenzi, Accorsi et al. (2021) https://doi.org/10.3389/fpsyt.2021.716488

• Provenzi, Grumi, Altieri, et al. (2021) https://doi.org/10.1017/S0954579421000766

• Provenzi, Mambretti, Villa, et al (2021) https://doi.org/10.1038/s41598-021-95053-z

• Roberti, Giacchero, Grumi et al. (2022) https://doi.org/10.1007/s10995-022-03540-0

• Nazzari, Grumi, Mambretti et al. (2022) https://doi.org/10.1038/s41398-022-02160-0

• Provenzi, Villa, Mambretti et al. (2022) https://doi.org/10.3389/fpsyt.2022.950455

A. Physical exposure (response: Yes, No)

During pregnancy…

1 I tested positive for COVID-19

2 I had symptoms reminiscent of COVID-19

3 I had contacts with relative or friends who tested positive for COVID-19

4 I live in a high contagion zone (e.g., red zone)

5 I had contacts with relatives or friends who live in a high contagion zone (e.g., red zone)

6 One of my relatives or friends was hospitalized due to COVID-19 infection

7 One of my relative or friends died with COVID-19

B. Pandemic related stress (Response: 5-point Liker scale)

During pregnancy…

1= not at all; 2= slightly; 3= Moderately; 4= Very much; 5= Extremely;

1 How much worried were you about the risk of COVID-19 infection?

2 How much did you feel that your pregnancy was at risk due to COVID-19 pandemic?

3 How much did you fear for your health?

4 How much did you fear for your baby's health?

5 How much did you feel that you were losing confidence in your health?

6 How much did you feel you had lost faith in medicine?

Intro

1) The authors cite evidence related to increases in depression during the pandemic, but the main predictor in this paper is perceived stress. Clinical symptoms of mental health disorders are different from perceived stress levels, and we should avoid conflating the two constructs.

REPLY. To avoid misunderstanding, we removed from the introduction section references about depression and replaced them with evidence about prenatal maternal distress during the pandemic (Yan, Ding, & Guo, 2020).

Method

2) Need more information about how the domains of socio-cognitive functioning were coded. Were there multiple coders, and how was reliability assessed?

REPLY. We thank the Reviewer for highlighting that more information on the coding procedure was needed. Two main coders were trained on the coding procedure described in the manuscript by the PI, were blind to all other study data and rated the videos independently. A subset of randomly selected videos (approximately 10%) were independently coded by two independent coders and the percentage of inter-rater agreement was 99.48%. We have added these details in the text.

3) The authors mention there was “remarkable sample attrition,” thus they only report results on infants with complete data. What was the original sample size, and what are the patterns of missing data? It is likely that large amounts of attrition will bias results. The authors should also consider using some method of imputation to reduce bias.

REPLY. The initial sample size at childbirth was n= 320 mother-child dyads, but only n=91 dyads took part in the observational task planned at 12 months, with an attrition rate of 71.56% (we added this details in the text). All infants were full-term births without clinical conditions. Dyads of the final sample of the present study did not significantly differ from those who dropped out in terms of stress experienced during pregnancy and socio-demographic characteristics, except for gestational age (M= 39.90; SD 1.05 for the final sample; M= 39.54, SD 1.09 for the participants who dropped out) and maternal education (M= 15; SD 3.35 for the final sample; M= 15.86, SD 2.91 for the participants who dropped out):

• Pandemic related stress t(318)= -.489, p=.625

• Gestational age t(318)= -2.687, p=.008

• Birth weight t(318)= -1.123, p= .262

• Infants’ sex χ2(1)= .053, p= .817

• Apgar score (min 5)= t(317)= -.232, p= .816

• Maternal age t(316)= .418, p=.676

• Maternal education t(317)= -2.124, p= .034

• Maternal job χ2(1)= .002, p= .967

These details have now been included in the methods section and acknowledged in the limitations. Lastly, we did not consider using any methods of imputation given that among dyads who took part in the observational task there was no missing data.

4) Limitations of the remote interaction task should be considered. Are there any other published studies that have coded a remote parent-child interaction task? This is a novel method, and any evidence linking remote recordings to in-person recordings should be cited. If not, this is an important limitation to acknowledge, as the home environment is different from a lab environment (and, there might be connectivity issues with coding interactions remotely, etc). Additionally, I would appreciate more details on the remote interaction (overall quality of the recordings, percent of recordings that were unusable, instructions given to the parent, etc) as this may be of use to future researchers.

REPLY. The reviewer is right, this is a novel method that implies specific strengths and limitations. It allowed us to collect observational data during the pandemic, but it will be helpful also in the post-pandemic period to maximize the participation in longitudinal studies for families living far from labs/hospitals. Moreover, it allows to collect more ecological data, observing the infants’ behaviors in a context that is familiar for them. However, it includes also specific challenges to take care of as: verifying the quality of internet connection; engaging the mother in positioning the smartphone/tablet in the right position to have the full side-view of both the interactive partners; giving instructions to the mother to set the smartphone/tablet with a black background so as not to distract the child. 

Overall, our procedure was consistent with recommendations by Shin et al. (2021) doi: 10.3389/fpsyg.2021.703822 for behavioral remote data collection in the home setting. And this allowed us to collect a high number of video suitable for coding. Indeed, a total of 102 tasks were recorded, but only 91 were adapt for coding and were used for the present study. All videos had good output quality which made them suitable for coding. A total of 11 videos were removed from the study because the task was not performed as expected: in most cases the infant was crying in the highchair and was held by the mother for part of the task; in one case another child entered in the room during the recording.

A validation study of this remote auditory task is being submitted to another journal. We have added details in the manuscript and presented limitations in the discussion section.

Results

5) An eligibility criteria for the study required mothers to test negative for COVID at delivery. The authors cite important evidence by Shuffrey et al, which reports neurodevelopmental delays of infants born to mothers who reported a COVID infection during pregnancy. Therefore, the authors should conduct a robustness check on mothers who tested positive for COVID during pregnancy with infant SCS scores (frequency & rate) if possible as a supplemental analysis.

REPLY. We think that there was a misunderstanding regarding our reference to Shuffrey’s evidence: his results suggested that birth during the pandemic, but not in utero exposure to maternal SARS-CoV-2 infection, was associated with differences in neurodevelopment at age 6 months. Unfortunately, we cannot try to replicate this result in our study about early socio-cognitive development. Indeed, as now specified also in the manuscript, in our final sample only two mothers reported testing positive to COVID-19 during pregnancy. Thus, it is not possible to perform supplemental analysis to control the possible specific influence of the COVID infection. However, as the current results are based on a sample of infants not exposed to the infection during pregnancy, strengthen the hypothesis that prenatal pandemic-related stress per se, rather than the infection, may be associated to the infants’ early socio-cognitive development.

6) The authors collected data on not just the occurrence, but also the frequency by which infants displayed these pointing/communication behaviors during the auditory task. These results are missing from the manuscript

REPLY. The descriptive data for the dichotomous occurrence (no = 0, yes = 1) is reported at the beginning of the Results section. We have now included a Figure (new Figure 1) reporting the distribution of the continuous (frequency) measure of pointing and communication. We agree with the reviewer that this adds to the clarity of the manuscript.

7) The authors test for correlations between potential confounds but do not report the results of these tests. Why not just control for confounds using linear regression? Even if the covariates are not significantly related to either the predictor or outcome variable, they are theoretically important to the outcome and accounting for them will result in a more precise estimate. I recommend the authors use logistic regression for the dichotomous indicators to obtain odds ratios (and linear regression with standardized predictor variables for the continuous outcomes), and include covariates for the demographic characteristics listed

REPLY. We thank the Reviewer for this suggestion. We have now included separate hierarchical regression models for each of socio-cognitive outcomes (i.e., SCS, PC-ORI, PC-POI, PC-COM), adjusting for potential infants (step 1) and maternal (step 2) confounders. We have adjusted the discussion accordingly. Also we revised all the Figures to make them clearer for the reader.

8) Did you account for age of the infant at time of the assessment? This should also be included as a covariate, as there might be differences in developmental timing of pointing/communication abilities by age.

REPLY. All infants were tested at 12 months ± 14 days. Specifically, mean age at testing was 376 (sd = 10, min = 332, max = 400): the youngest infant was 10 months and 27 days old, whereas the oldest was 1 year, 1 month and 3 days old. As such, considering the very narrow age range we did not expect any significant effect of age on the observed outcomes. 

9) What is the socioeconomic spread of the sample? Maternal education is reported, but is there any further information such as income?

REPLY. Unfortunately we did not have any detailed information about the parents’ income. While we did not have any precise assessment of the socioeconomic status of the participants, we can confirm that the sample was a low-risk community sample.

10) In observing Figure 2, there appears to be an outlier (high PC-COM score). Are these results robust to outliers? This should be reported in the supplement, given the small sample

REPLY. We excluded the outlier and repeated the analyses. Findings – in terms of statistical significance – did not differ, so we keep the complete set of 91 dyads.

Discussion:

11) The discussion begins with the sentence “the present study is among the first to suggest the presence of a long-lasting effect of maternal stress…”. This terminology is again used in the next paragraph “…socio-cognitive behaviors appeared to be at least partially affected by maternal stress exposure during pregnancy.” This implies causation.

REPLY. We agree with the reviewer about the fact that the longitudinal nature of our data is no guarantee for drawing valid causal inferences, excluding all competing explanations. We rephrased these sentences in terms of association.

12) This study would be helpful to cite in either the intro or discussion – it is not correlational, which provides more compelling evidence: Imboden, A., Sobczak, B. K., & Griffin, V. (2021). The impact of the COVID-19 pandemic on infant and toddler development. Journal of the American Association of Nurse Practitioners. https://doi.org/10.1097/JXX.0000000000000653

REPLY. Thank you for the suggestion, we have cited this study both in the introduction and discussion.

13) This recent study also is a natural experiment: Sperber, J. F., Hart, E. R., Troller-Renfree, S. V., Watts, T. W., & Noble, K. G. (2022). The effect of the COVID-19 pandemic on infant development and maternal mental health in the first 2 years of life. Infancy. https://doi.org/10.1111/infa.12511

REPLY. Thank you for the suggestion, we have cited this study in the introduction.

14) I challenge the authors to remove terminology such as “suboptimal” development, as it assigns value judgements to developmental trajectories. It is possible (and there is a large body of literature arguing this) that developmental differences may be “suboptimal” according to traditional Western contexts, but may be adaptive and advantageous in adverse contexts. I encourage the writers to remove value judgements of developmental trajectories, and explore this idea further – how might socio-cognitive difference due to early stress exposure be adaptive in certain contexts?

REPLY. We thank the reviewer for this input, we have rephrased that terminology accordingly.

---

## [Decision Letter · Decision Letter 1]

17 Feb 2023

PONE-D-22-23782R1Maternal pandemic-related stress during pregnancy associates with infants’ socio-cognitive development at 12 months: A longitudinal multi-centric studyPLOS ONE

Dear Dr. Provenzi,

Thank you for submitting your manuscript to PLOS ONE. After careful consideration, we feel that it has merit but does not fully meet PLOS ONE’s publication criteria as it currently stands. Therefore, we invite you to submit a revised version of the manuscript that addresses the points raised during the review process.

We look forward to receiving your revised manuscript.

Kind regards,

Claudine Irles, Ph.D.

Academic Editor

PLOS ONE

Journal Requirements:

Reviewers' comments:

Reviewer's Responses to Questions

**Comments to the Author**

1. If the authors have adequately addressed your comments raised in a previous round of review and you feel that this manuscript is now acceptable for publication, you may indicate that here to bypass the “Comments to the Author” section, enter your conflict of interest statement in the “Confidential to Editor” section, and submit your "Accept" recommendation.

Reviewer #2: All comments have been addressed

Reviewer #3: (No Response)

2. Is the manuscript technically sound, and do the data support the conclusions?

Reviewer #2: Yes

Reviewer #3: Partly

3. Has the statistical analysis been performed appropriately and rigorously? 

Reviewer #2: Yes

Reviewer #3: No

4. Have the authors made all data underlying the findings in their manuscript fully available?

Reviewer #2: Yes

Reviewer #3: Yes

5. Is the manuscript presented in an intelligible fashion and written in standard English?

Reviewer #2: Yes

Reviewer #3: Yes

6. Review Comments to the Author

Reviewer #2: The authors did an excellent job responding to reviewer concerns. The current manuscript is well-written, presents compelling evidence, and will make a valuable contribution to the literature. I am happy to recommend this paper for publication. A few small notes:

Though statistically significant differences were found in maternal education and gestational age between those who attritted from the study and those who participated in the 12-month assessment, the magnitude of this difference is small (and ultimately, inconsequential). I think the authors should acknowledge this and present it as such using the mean differences as evidence. Rather, the bigger concern is that there are unobserved difference between the hundreds of dyads that attritted from the study and the follow-up sample (this is quite likely). I request the authors raise this concern in the limitation section, rather than the statistically significant differences found in the t-test.

Small typo found on the first line of page 11: “thought” should be “though”

Figure 4 is a bit overwhelming to the eye. I recommend adding sub-titles above each scatterplot regarding which infant outcome is being represented (rather than only depending on the Y-axis) and limiting only 2 infant outcomes to a page. Additionally, I recommend putting the standardized coefficient and p-value from the regression analysis under each figure to contextualize the figure for the reader. Also specify for all figures when variables are standardized.

Reviewer #3: This is a highly interesting and relevant manuscript relating pandemic related stress to socio-cognitive development in 12 month old infants. Many of the comments raised in the previous version have been well addressed. As I did not perform the previous review I have a few relatively minor comments.

1. The authors argue that they found a "fairly lnear or dose response relationship...", however, the regression models that they ran did not really test whether the relationships are linear. To do this the investigators would need to evaluate either a quadratic term for the PRS measure or perform spline analyses. This is important because the shape of the dose response curve matters when policy makers are considering the availability of 'formal and informal' support (although such support is always good). Indeed, inspecting figure 4 leads me to conjecture that (a) for some of the outcomes such as SCS and PC-ORI the relationships are not linear and (b) the relationships may differ for male and female infants.

2. The large degree of missing data (ie loss to follow up) is troubling, especially since follow up is related to important confounders such as maternal education and gestational age. This could be better handled by performing inverse probability weighting which would account for some of the potential bias due to differential loss to follow up.

3. Please use more descriptive headers for the tables. For example, in tables 1 and 2 should say something about the sample size, and whether this was an exploratory or confirmatory PCA. Table 3 should also have the sample size and indicate if this is the larger sample or the analytic sample. further this table should give the data to evaluate the loss to follow up, i.e. compare those who were followed at 12 months to those who were not.

4. Please indicate whether successive steps include the variables in the previous step. That would make sense, but it is not clear the way the table is formatted.

5. Because the scores on the socio-cognitive development measures were not normally distributed there is a chance that extreme outliers may influence the results. Did the authors also consider the log transform of these variables, which would normalize the distribution.

7. PLOS authors have the option to publish the peer review history of their article (what does this mean?). If published, this will include your full peer review and any attached files.

Reviewer #2: No

Reviewer #3: No

---

## [Author Response · Author response to Decision Letter 1]

7 Mar 2023

Article ID: PONE-D-22-23782R1

Title: Maternal pandemic-related stress during pregnancy associates with infants’ socio-cognitive development at 12 months: A longitudinal multi-centric study

Replies to comments from reviewers

Reviewer #2

1. The authors did an excellent job responding to reviewer concerns. The current manuscript is well-written, presents compelling evidence, and will make a valuable contribution to the literature. I am happy to recommend this paper for publication.

Reply. Thanks for appreciating our revision.

2. A few small notes: Though statistically significant differences were found in maternal education and gestational age between those who attritted from the study and those who participated in the 12-month assessment, the magnitude of this difference is small (and ultimately, inconsequential). I think the authors should acknowledge this and present it as such using the mean differences as evidence. Rather, the bigger concern is that there are unobserved difference between the hundreds of dyads that attritted from the study and the follow-up sample (this is quite likely). I request the authors raise this concern in the limitation section, rather than the statistically significant differences found in the t-test.

Reply. Thanks for this suggestion, we have modified the text as follows:

“Additionally, potential systematic patterns of missingness limit generalizability of the current findings. In particular, while small differences in gestational age and maternal education were found among dyads who attended the follow-up and dyads who withdrew, only variables collected at birth were available for comparison among the two groups, thus possible selective attrition on unobserved variables remained an issue.”

3. Small typo found on the first line of page 11: “thought” should be “though”

Reply. Thank you again for careful reading of our manuscript. We have fixed the typo. 

4. Figure 4 is a bit overwhelming to the eye. I recommend adding sub-titles above each scatterplot regarding which infant outcome is being represented (rather than only depending on the Y-axis) and limiting only 2 infant outcomes to a page. Additionally, I recommend putting the standardized coefficient and p-value from the regression analysis under each figure to contextualize the figure for the reader. Also specify for all figures when variables are standardized.

Reply. We agree with Rev #2. Figure 4 has been split into two new figures: the new Fig 4 reports the scatterplots for ORI, POI, COM; the new Fig 5 reports the scatterplot for the cumulative SCS. Statistics are reported in the text – anyway, they are also replicated in the Table legend now. These legends also highlight that all outcome variables on the y-axis are standardized.

Reviewer #3

This is a highly interesting and relevant manuscript relating pandemic related stress to socio-cognitive development in 12 month old infants. Many of the comments raised in the previous version have been well addressed. As I did not perform the previous review I have a few relatively minor comments.

Reply. Thanks for appreciating our work.

1. The authors argue that they found a "fairly linear or dose response relationship...", however, the regression models that they ran did not really test whether the relationships are linear. To do this the investigators would need to evaluate either a quadratic term for the PRS measure or perform spline analyses. This is important because the shape of the dose response curve matters when policy makers are considering the availability of 'formal and informal' support (although such support is always good). Indeed, inspecting figure 4 leads me to conjecture that (a) for some of the outcomes such as SCS and PC-ORI the relationships are not linear and (b) the relationships may differ for male and female infants.

Reply. We thank the Reviewer for making this point. It should be highlighted that the sentence “fairly linear or dose response relationship” was not referred to our findings, but to previous research – nonetheless, it has now been removed to avoid misinterpretations. At the same time, non-linearity of the observed associations was examined by the inclusion of a quadratic term for PRS within the regression models. Results for a quadratic effect of PRS on the study outcomes are as follows:

• No significant effect for ORI (as for linear testing)

• No significant effect for POI (as for linear testing)

• A tendency to statistical significancy for COM (R-squared = .10, p = .059; while the PRS linear term yielded a significant effect)

• A significant effect for SCS (R-squared = .09, p = .027; the linear effect had R-squared = .10, p = .017).

As such, we did not change our analytical plan. Nonetheless, adding the comparison between the linear and the quadratic association was not within the aims of the paper and we do not traced in previous literature enough support to make specific hypotheses on the shape of the linear or curvilinear association. Consistently, we opted to report on this linear association as it was tested as linear – of course, this does not rule out that in larger populations or in different settings the association might be curvilinear. For instance, few studies provided evidence that the association between maternal prenatal stress and children’s outcomes might be U-shaped, with mild to moderate levels of maternal distress during pregnancy actually promoting child development in healthy samples and being associated with more optimal outcomes, as compared to too little or too much prenatal maternal stress (e.g., Fernandes et al., 2014; DiPietro et al., 2006; Davis et al., 2017).

2. The large degree of missing data (ie loss to follow up) is troubling, especially since follow up is related to important confounders such as maternal education and gestational age. This could be better handled by performing inverse probability weighting which would account for some of the potential bias due to differential loss to follow up.

Reply. We thank the Reviewer for raising this issue. We agree that sample attrition is actually a significant issue in the current sample and is probably related to the pandemic time when recruitment occurs that limited participants’ engagement by allowing only remote data collection. Unfortunately, we only have information collected at birth about the dyads who withdrew from the study and, as also highlighted by Reviewer 2, while small differences were detected in terms of gestational age and maternal education among participants and not-participants to the follow-up phases, a number of unobserved differences are likely to exist, thus making our sample not representative of all observations. Thus, we do not feel appropriate to use inverse probability weighing in the current sample, as its effectiveness in “correcting” the selection bias highly depends on the availability of enough information, for the entire population, to predict the non-missingness probability (Oris et al., 2016; Narduzzi et al., 2014). 

Following also comments from Reviewer 2, this has now been better acknowledged in the limitation section. 

Oris, M., Roberts, C., Joye, D., & Stähli, M. E. (2016). Surveying human vulnerabilities across the life course: Balancing substantive and methodological challenges. In M. Oris, C. Roberts, D. Joye, & M. E. Stähli (Eds.), Surveying human vulnerabilities across the life course (pp. 1–25). Springer International Publishing. https://doi.org/10.1007/978-3-319-24157-9_1

Narduzzi, S., Golini, M. N., Porta, D., Stafoggia, M., & Forastiere, F. (2014). L'uso dell'Inverse probability weighting (IPW) nella valutazione e "correzione" del selection bias [Inverse probability weighting (IPW) for evaluating and "correcting" selection bias]. Epidemiologia e prevenzione, 38(5), 335–341.

3. Please use more descriptive headers for the tables. For example, in tables 1 and 2 should say something about the sample size, and whether this was an exploratory or confirmatory PCA. Table 3 should also have the sample size and indicate if this is the larger sample or the analytic sample. further this table should give the data to evaluate the loss to follow up, i.e. compare those who were followed at 12 months to those who were not.

Reply. Sample size has now been reported in all Table headers. Additionally, sociodemographic characteristics of the sample of dyads who dropped out from the study as well as differences with the study sample has now been summarized in Supplementary Table 1.

4. Please indicate whether successive steps include the variables in the previous step. That would make sense, but it is not clear the way the table is formatted.

Reply. Yes, the reviewer is correct. This has been added in the text.

5. Because the scores on the socio-cognitive development measures were not normally distributed there is a chance that extreme outliers may influence the results. Did the authors also consider the log transform of these variables, which would normalize the distribution.

Reply. This is a relevant issue. We indeed discussed about the possibility or opportunity to normalize the distribution. Nonetheless, it should be considered that these measures refer to specific precursors of socio-cognitive development that are emerging at the age (12-month) at which they were tested in this study. As such, it is expected that their distribution would be asymmetric as they are, reflecting the fact that some infants have already “mastered” these skills, while others are moving toward the same achievement. The distribution of these measures is consistent with this expectation. Indeed, we report here the asymmetry and kurtosis indexes for the socio-cognitive development measures:

Measure Asymmetry Kurtosis

ORI 0.41 -0.27

POI 2.26 4.94

COM 2.21 7.18

SCS 1.21 1.44

If we log-transform these measures, we obtain:

Measure Asymmetry Kurtosis

ORI 0.18 -0.37

POI 2.04 3.84

COM 1.72 4.27

SCS 0.95 0.78

These indexes are still problematic for some of the measures. Anyway, we ran the analyses using the log-transformed measures and we obtained comparable findings (we report here the final regression step):

• ORI: statistically non-significant

• COM: adjusted R-squared = .10, p = .042; PRS coefficient = -2.06, p = .042

• POI: statistically non-significant

• SCS: adjusted R-squared = .10, p = .016; PRS coefficient = -2.45, p = .016

In the light of these findings, we opted not to change our manuscript introducing the log-transformation.

---

## [Editor Report · Decision Letter 2]

4 Apr 2023

Maternal pandemic-related stress during pregnancy associates with infants’ socio-cognitive development at 12 months: A longitudinal multi-centric study

PONE-D-22-23782R2

Dear Dr. Provenzi,

We’re pleased to inform you that your manuscript has been judged scientifically suitable for publication and will be formally accepted for publication once it meets all outstanding technical requirements.

Kind regards,

Claudine Irles, Ph.D.

Academic Editor

PLOS ONE
---

## [Editor Report · Acceptance letter]

6 Apr 2023

PONE-D-22-23782R2 

Maternal pandemic-related stress during pregnancy associates with infants’ socio-cognitive development at 12 months: A longitudinal multi-centric study 

Dear Dr. Provenzi:

I'm pleased to inform you that your manuscript has been deemed suitable for publication in PLOS ONE. Congratulations! Your manuscript is now with our production department. 

Kind regards, 

on behalf of

Dr. Claudine Irles 

Academic Editor

PLOS ONE